# Stable and bright formamidinium-based perovskite light-emitting diodes with high energy conversion efficiency

Yanfeng Miao [1], You Ke[1], Nana Wang [1], Wei Zou[1], Mengmeng Xu[1], Yu Cao [1], Yan Sun[1], Rong Yang[1], Ying Wang[1], Yunfang Tong[1], Wenjie Xu[1], Liangdong Zhang[1], Renzhi Li[1], Jing Li[2], Haiping He [2], Yizheng Jin [3], Feng Gao [4], Wei Huang[1,5,6] & Jianpu Wang [1]

Solution-processable perovskites show highly emissive and good charge transport, making them attractive for low-cost light-emitting diodes (LEDs) with high energy conversion efficiencies. Despite recent advances in device efficiency, the stability of perovskite LEDs is still a major obstacle. Here, we demonstrate stable and bright perovskite LEDs with high energy conversion efficiencies by optimizing formamidinium lead iodide films. Our LEDs show an energy conversion efficiency of 10.7%, and an external quantum efficiency of 14.2% without outcoupling enhancement through controlling the concentration of the precursor solutions. The device shows low efficiency droop, i.e. 8.3% energy conversion efficiency and 14.0% external quantum efficiency at a current density of 300 mA cm$^{-2}$, making the device more efficient than state-of-the-art organic and quantum-dot LEDs at high current densities. Furthermore, the half-lifetime of device with benzylamine treatment is 23.7 hr under a current density of 100 mA cm$^{-2}$, comparable to the lifetime of near-infrared organic LEDs.

[1] Key Laboratory of Flexible Electronics (KLOFE) & Institute of Advanced Materials (IAM), Jiangsu National Synergetic Innovation Center for Advanced Materials (SICAM), Nanjing Tech University (NanjingTech), 30 South Puzhu Road, 211816 Nanjing, China. [2] State Key Laboratory of Silicon Materials, School of Materials Science and Engineering, Zhejiang University, 310027 Hangzhou, China. [3] Center for Chemistry of High-Performance and Novel Materials, State Key Laboratory of Silicon Materials, and Department of Chemistry, Zhejiang University, 310027 Hangzhou, China. [4] Biomolecular and Organic Electronics, IFM, Linköping University, 58183 Linköping, Sweden. [5] Key Laboratory for Organic Electronics and Information Displays & Institute of Advanced Materials (IAM), Jiangsu National Synergetic Innovation Center for Advanced Materials (SICAM), Nanjing University of Posts & Telecommunications, 9 Wenyuan Road, 210023 Nanjing, China. [6] Shaanxi Institute of Flexible Electronics (SIFE), Northwestern Polytechnical University (NPU), 127 West Youyi Road, 710072 Xi'an, China. Correspondence and requests for materials should be addressed to W.H. (email: iamwhuang@nwpu.edu.cn) or to J.W. (email: iamjpwang@njtech.edu.cn)

In general, high-performance perovskite light-emitting diodes (PeLEDs) require high-quality perovskite thin films with good emission properties and complete surface coverage to minimize nonradiative recombination. Methylammonium (MA) based three-dimensional (3D) perovskites are found difficult to achieve thin films with complete coverage, although various interfacial approaches have been applied[1–4]. By using around 400 nm perovskite film to improve the coverage, Cho et al. has obtained PeLED with an external quantum efficiency (EQE) of 8.3%[5]. However, such a thick perovskite film can reduce charge transport and increase operation voltage of the devices, which leads to low energy conversion efficiencies (ECEs). Alternatively, multiple-quantum-well (MQW) perovskites exhibit excellent film uniformity and emission properties, resulting in a high electroluminescence (EL) EQE of 11.7%, a good ECE of 5.5% at 100 mA cm$^{-2}$ and improved operation stability[6]. Similar strategy was also independently developed by Yuan et al.[7]. Recently, by using additive or insulating polymers to passivate defects or form microstructures to enhance the light outcoupling, the EQEs of PeLEDs have reached 20%[8–11]. However, the ECE of these PeLEDs can be potentially limited by unfavorable charge transport caused by additives or polymers incorporated in perovskite films[8–12]. In principle, in order to improve the LED operation stability, high ECE is required to reduce the thermal energy generated in the device[13]. Therefore, ideal perovskite thin films for LED applications must have the synergistically integrated merits of high PLQEs and good charge transport, which are currently lacking. Here we demonstrate the fabrication of such perovskite films based on pure FAPbI$_3$ by simply controlling the concentration of the precursor solutions during spin-coating process.

## Results

### Stable and bright PeLEDs with high ECEs.

Figure 1a shows the multilayer structure of our devices, consisting of indium tin oxide (ITO)/polyethylenimine ethoxylated (PEIE)-modified zinc oxide (ZnO, 30 nm)/FAPbI$_3$ perovskite (50 nm)/poly(9,9-dioctyl-fluorene-co-N-(4-butylphenyl)diphenylamine) (TFB, 40 nm)/molybdenum oxide (MoO$_x$, 7 nm)/gold (Au, 60 nm). Here the ZnO/PEIE and TFB are electron transport layer (ETL) and hole transport layer (HTL), respectively[6]. The FAPbI$_3$ perovskite films were deposited by using an anti-solvent method[14], otherwise the perovskite film has a very low surface coverage (Supplementary Fig. 1). The precursor solution consists of a 10 wt.% of formamidinium iodide (FAI) and PbI$_2$ with a molar ratio of 2:1 in N,N-dimethylformamide (DMF). We note that the concentration of the precursor solution is critical, which will be discussed in the next section.

The EL emission peak of the fabricated device is located at 804 nm (Fig. 1b), which is slightly blue-shifted compared to the EL emission of FAPbI$_3$ LEDs fabricated without using the anti-solvent method[6]. The shape of the EL spectra remains unchanged upon various biases (Supplementary Fig. 2). Figure 1c shows the current density-radiance-voltage characteristics of the PeLED. The device has a low turn-on voltage of 1.3 V. The current density and EL increase rapidly over the bias voltage, indicating good charge transport of the 3D perovskite film. At a low voltage of 2.75 V, a brightness up to 241 W sr$^{-1}$ m$^{-2}$ can be achieved. The angular emission intensity of the device follows a Lambertian profile (Supplementary Fig. 2). Figure 1d shows that the EQE of the device reaches up to 14.2% at a high current density of 188 mA cm$^{-2}$ and a high brightness of 131.2 W sr$^{-1}$ m$^{-2}$. Due to the high EQE and low operation voltage, this LED exhibits a high ECE of 10.3% at a current density of 100 mA cm$^{-2}$ (corresponding to a brightness of 67 W sr$^{-1}$ m$^{-2}$). We note that in literature the highest ECE at 100 mA cm$^{-2}$ without outcoupling enhancement was 9.2, 5.6, and 5.5% for PeLEDs[15], organic LEDs[16], and quantum-dot LEDs[17], respectively. Even at a high current density of 300 mA cm$^{-2}$, our device shows an EQE of 14.0% and an ECE of 8.3%, suggesting exceptional low efficiency droop. The statistics

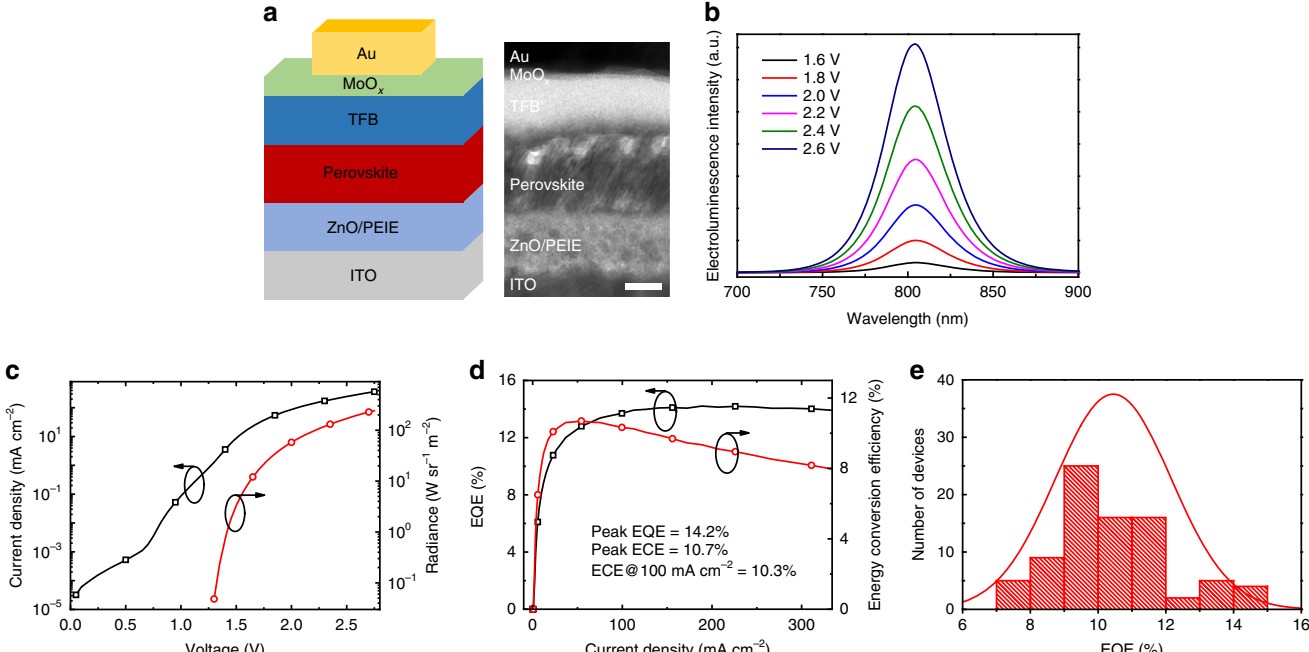

**Fig. 1** PeLED device structure and optoelectronic characteristics. **a** Device structure and cross-sectional HRTEM image (scale bar, 20 nm). **b** Device EL spectra upon various biases. **c** Current density and radiance versus driving voltage for the highest EQE device (10 wt.%). Radiance of 241 W sr$^{-1}$ m$^{-2}$ is obtained under 2.75 V. **d** EQE and ECE versus current density for the highest EQE device (10 wt.%). A peak EQE of 14.2% is achieved at a current density of 188 mA cm$^{-2}$ and a peak ECE of 10.7% is obtained at a current density of 54 mA cm$^{-2}$. **e** Histogram of peak EQEs measured from 82 devices, which shows an average peak EQE of 10.5% with a relative standard deviation of 16%

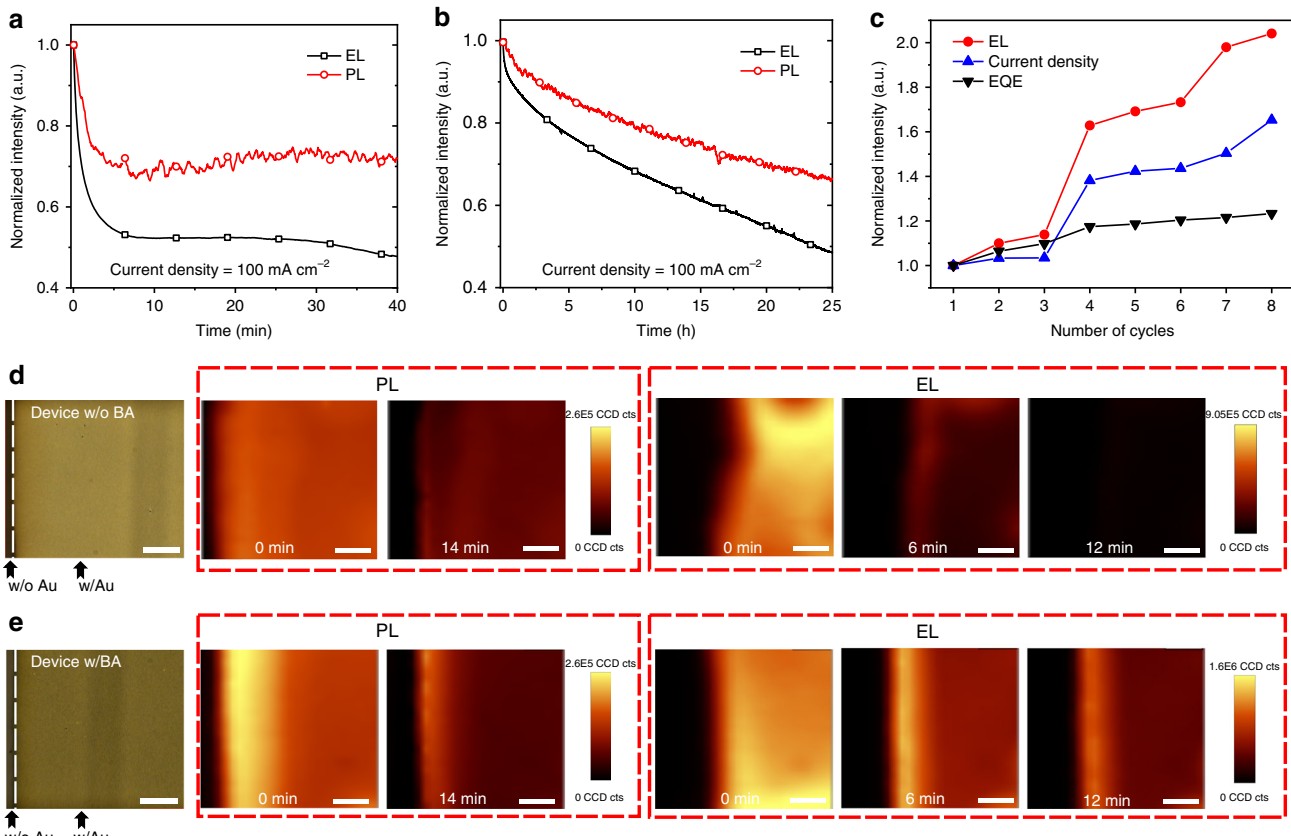

**Fig. 2** Optoelectronic characteristics of the aged devices. **a** EL and PL decay of the 3D PeLEDs without BA treatment at a constant current density of 100 mA cm$^{-2}$. **b** EL and PL decay of the 3D PeLEDs with BA treatment at a constant current density of 100 mA cm$^{-2}$. **c** Normalized device EL, current density and EQE under cyclic voltages between −1 and 2.5 V after aging test. The EL/current density/EQE show recovery. **d** Microscopy image and corresponding PL and EL intensity maps of device without BA treatment (scale bar, 70 μm), which show the degradation starts from the edge of Au electrode and the EL degrades faster than the PL. **e** Microscopy image and corresponding PL and EL intensity maps of device with BA treatment (scale bar, 70 μm), which show that the BA interfacial layer significantly suppresses the device degradation

of 82 devices shows an average peak EQE of 10.5% with a relative standard deviation of 16% (Fig. 1e), suggesting that the device performance is reasonably reproducible.

We highlight that our PeLEDs with high ECEs show remarkable operation stability. A device with simple glass-epoxy encapsulation method, exhibits a lifetime ($T_{50}$, time to half of the initial brightness) of 0.5 h under a large constant current density of 100 mA cm$^{-2}$ (Fig. 2a). In contrast, previously reported pure FAPbI$_3$ perovskite LED can only survive a few minutes at a much lower current density (10 mA cm$^{-2}$)[6]. We believe that the significantly improved stability is due to the high ECE, since low efficiency FAPbI$_3$ PeLED devices (with 20 wt.% of precursor solution) can only have similar lifetime to previous results. We have measured the stability of FAPbI$_3$ LEDs under 25, 35, and 45 °C in a glovebox. The result shows that the stability is strongly dependent on the temperature (Supplementary Fig. 3), indicating the high ECE can effectively improve the stability due to less thermal energy generated. We find the device stability can be further enhanced by benzylamine (BA) treatment which was previously demonstrated to be effective as a moisture-resistance and defect-passivation to suppress ion migration in perovskite solar cells[18,19]. After BA treatment, the photoluminescence (PL) emission intensity was significantly enhanced which can be due to the further reduced defects in grain boundary or surface of the perovskite film (Supplementary Fig. 4). And the X-ray diffraction (XRD) pattern is almost identical to the untreated film, showing α-phase FAPbI$_3$ (Supplementary Fig. 4). The device $T_{50}$ is increased to 23.7 h under a constant current density

of 100 mA cm$^{-2}$ (Fig. 2b). This result indicates a more than 3 orders of magnitude enhancement compared to previous pure FAPbI$_3$ based PeLEDs[6]. For reference, under much lower current density (10 mA cm$^{-2}$), the reported $T_{50}$ of state-of-the-art near-infrared OLED is 60 h (Supplementary Table 1)[20]. We note that the peak EQE of the device with BA treatment is slightly decreased, which is likely due to the increased roughness in perovskite films upon BA treatment (Supplementary Fig. 4).

The degraded EL can be caused by the degradation of either emission properties of the perovskites or the electrical transport properties of the devices. In order to reveal more detail on the aging process, we simultaneously measure the PL and EL by using chopped laser light and lock-in amplifier[21,22]. During the measurement, the device is under a constant current density of 100 mA cm$^{-2}$, and the excitation light intensity (425 nm CW laser) was kept extremely low to avoid disturbing EL, around 0.2 mW cm$^{-2}$. The in situ PL/EL measurements show that the PL decline is slower than the EL decline (Fig. 2b). When the EL decreased to half of its initial value, the PL intensity was 67% of its initial value. We hypothesis that in addition to the degradation of the emission layer, the EL degradation is partially due to the deterioration of interfacial (or charge balance) properties of the devices, since the EL degradation is more seriously than the PL degradation. In order to verify this hypothesis, we measured EL and current density of an aged device under cyclic voltages between −1 and 2.5 V (Supplementary Fig. 5). The results show that the EL and current density can be recovered. After eight cycles, the EL intensity almost doubles

and the EQE is recovered by a factor of 1.23 at the bias voltage of 2.5 V (Fig. 2c). These results are consistent with the scenario that the interface is deteriorated during the aging process by formation of interfacial barrier, and the barrier can be reduced by using cyclic voltages[23,24]. The formation of bias-dependent interfacial barrier is likely associated with the ion migration in perovskites[25]. The observed recoverable EL performance is important for long-term application of PeLEDs.

In order to obtain more insights on the degradation process of our PeLEDs, we used a confocal fluorescence microscope to measure the PL and EL mapping of the devices in air (Fig. 2d, e). The PL and EL mapping measurements both show that the dark areas enlarge from the edge of metal electrode, indicating that degradation of devices starts from the edge, which is likely due to the fast penetration of moisture from the metal-uncovered part. Interestingly, the mapping measurement also shows that the EL dark area enlarges much more quickly than the PL dark area. In particular, for LED without BA treatment, there is almost no EL mapping signal after biasing for 12 min, while the PL intensity is still relatively high. This fact suggests the interfacial degradation plays important role in the aging process of PeLEDs, which is consistent with the above in situ PL/EL intensity measurement. Importantly, the BA treated device shows reduced degradation rate from the edge, and the EL and PL intensity in the center of device exhibits comparable rate of decrease. Since the BA treatment mainly improves the perovskite layer surface, we believe that the improved lifetime of device is mainly due to the reduced interfacial defects and enhanced moisture resistance of the perovskite film after BA treatment.

In addition to the 10 wt.% device, we also fabricated LEDs using perovskite films processed from precursor solutions with varied concentrations. Detailed device characterization results are presented in Supplementary Fig. 6. Briefly, the 7 wt.%-device and 15 wt.%-device show around 4% average peak EQE, and average peak EQEs of the rest two devices are below 2%. The statistics of the device EQEs shows that the device performance is in good reproducibility (Fig. 3f). The results suggest that the LED performance is highly dependent on the properties of the perovskite films.

**Properties of different concentration perovskite films**. The high-quality perovskite film processed from the 10 wt.% precursor solution is critical to achieve the exceptional LED performance. Scanning electron microscope (SEM) characterizations show full coverage of this film (Fig. 3), suggesting that leakage current due to direct contacting of ETLs and HTLs can be minimal. As shown in the Fig. 3, the film coverage becomes poor and discontinuous when the lower concentration solution is used. This leads to reduced EL efficiency due to the more morphological defects which can result in greater leakage current[2]. In addition, the PL lifetime of the films with lower concentration are 21 ns for 7 wt.% film and 17 ns for 5 wt.% film under a fluence of 4 nJ cm$^{-2}$ (Fig. 4a). The short PL lifetime can be attributed to the high defects presented in these poor perovskite films. In case of the higher concentration precursor solution than 10 wt.%, the perovskite film shows good coverage. However, we find that the optical properties of these perovskite films are much inferior to those of 10 wt.% film. Transient PL decay measurement shows that the PL lifetime of the 10 wt.% perovskite film is around 1 μs under a fluence of 4 nJ cm$^{-2}$ (Fig. 4a), indicating very low defect density. The PL lifetime is 68 and 59 ns for 15 wt.% and 20 wt.% films under the same fluence, respectively (Fig. 4a), indicating high trap densities presented in these high concentration films. Figure 4b also shows that PL lifetimes at various emission wavelengths are almost identical in the 10 wt.% perovskite film,

which suggests that the PL spectrum is homogenously broadened in the perovskite film[26]. This homogenously broadened PL emission can be observed in ordered system where the energy relaxation process is not important, such as high-quality vapor-deposited perovskite films[26]. In contrast, for most 3D perovskites with energetic disorder, transient PL decay at lower energy emission shows longer lifetime due to the energy migration process[27], which is consistent with the results of those higher concentration film (Supplementary Fig. 7).

The electronic disorder of perovskite films can be further quantified by using Fourier-transform photocurrent spectroscopy (FTPS) to determine the Urbach energy ($E_u$)[28]. Fig. 4c and Supplementary Fig. 8 show that the semi-log plot of incident photon-to-current efficiency (IPCE) absorption edge of the devices, which indicated an $E_u$ of 14.9 meV for the 10 wt.% perovskite film and 14.3 meV for the 10 wt.% perovskite film with BA treatment. We note these values are comparable to those previous reported $E_u$ for high-quality perovskite film (around 200 nm thickness)[29,30], although our perovskite film is only 50 nm, which must be more seriously affected by surface defects. A high PLQE of 45% can be observed with the 10 wt.% perovskite film at an excitation as low as 0.1 mW cm$^{-2}$, and the peak PLQE reaches up to 60% (Fig. 4d). The transient PL, PLQE and $E_u$ measurement results together suggest that the fabricated 10 wt.% FAPbI$_3$ perovskite thin film is highly energetically ordered with a very low level of defect density. While in regular 3D perovskite film, high PLQE can only be obtained at high excitation (higher than 100 mW cm$^{-2}$) when the trap-induced nonradiative recombination centers are filled[1,6]. The PLQEs of those higher concentration films are low (Fig. 4d), which is also consistent with the FTPS measurement (Supplementary Fig. 8). We have further measured the fluence dependent PL lifetime for the films with different precursor concentrations. The results show that all the films have decreasing lifetimes with increasing fluence and the 10 wt.% film has the longest lifetime at the low fluence due to suppressed trap-induced non-radiative recombination[31] (Supplementary Fig. 9). In addition, after BA treatment, the PL lifetime and PLQE of the 10 wt.% film are further increased (Fig. 4a, d), which is consistent with the passivation effect of the BA treatment.

In order to investigate the microscopic origin of different levels of disorder presented in those different perovskite films, cross-sectional samples of the perovskite films were analyzed by a high-resolution transmission electron microscopy (HRTEM). The HRTEM images show that the each grain observed in the SEM images (Fig. 3), consists of many small crystallites (Supplementary Fig. 10). The lattice fringe confirms that the small crystallites are α-phase FAPbI$_3$ perovskites (Supplementary Fig. 11)[32]. Moreover, Supplementary Fig. 10 also shows that the sizes of the crystallites vary in the different concentration perovskite films. The 10 wt.% perovskite has the largest crystal size. And the higher concentration perovskite films show smaller crystallites dispersed in the grains, consistent with the higher level of disorder presented. The different crystal sizes inside the grains are also consistent with the different extent of blue-shifted PL/EL emission peaks of the perovskite films (Supplementary Fig. 6d and Supplementary Fig. 12b). We think that the most blue-shifted PL/EL emission peak of the 20 wt.% films are likely due to the quantum-confinement effect of the small crystallites[33]. The largest crystallite with the 10 wt.% film has the weakest confinement effect, resulting in least blue-shifted PL/EL emission spectrum to 3D FAPbI$_3$ perovskite film fabricated by non-anti-solvent method (805 nm)[6]. Therefore, these results suggest that the high performance of 10 wt.% FAPbI$_3$ PeLED is due to both the full coverage of perovskite films consisting of grains and the

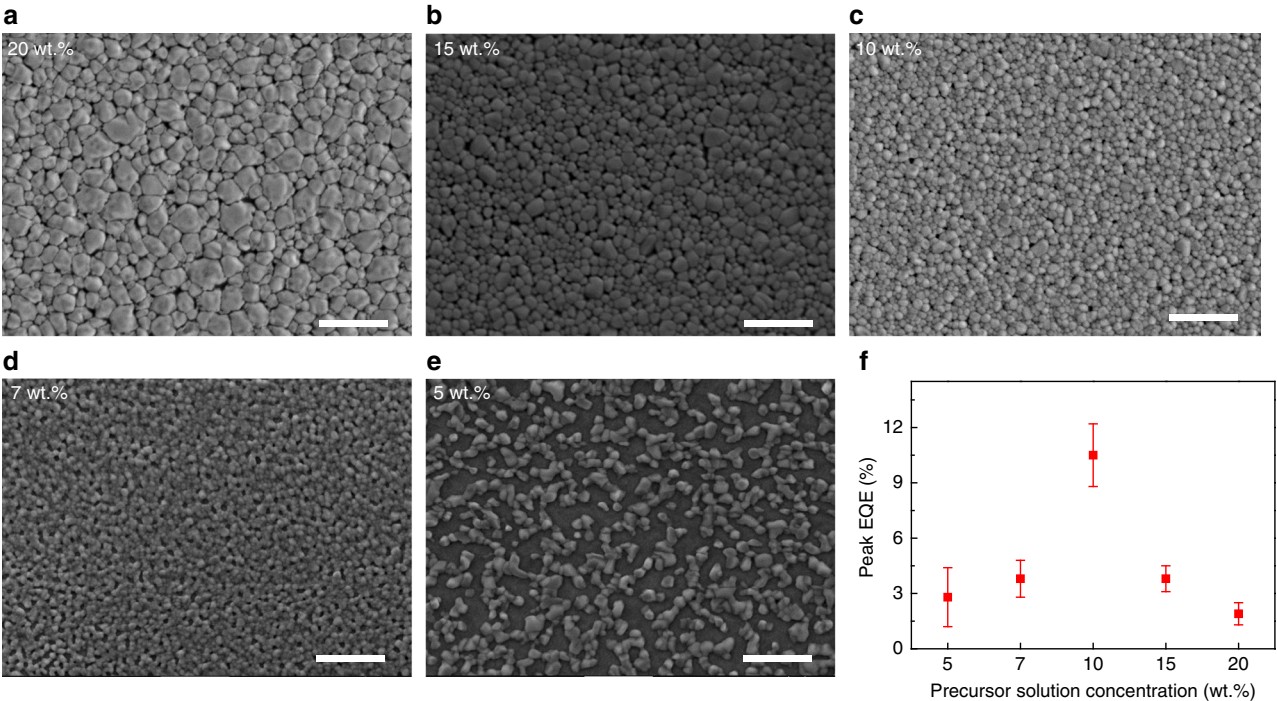

**Fig. 3** SEM images of FAPbI$_3$ films and EQE distribution of PeLEDs. Films and PeLEDs are fabricated with different concentrations of precursor solutions. **a** 20 wt.%. **b** 15 wt.%. **c** 10 wt.%. **d** 7 wt.%. **e** 5 wt.%. Scale bar, 1 μm. **f** Peak EQE distribution for devices with different concentrations. More than 20 devices for each concentration. Error bars correspond to the standard deviation

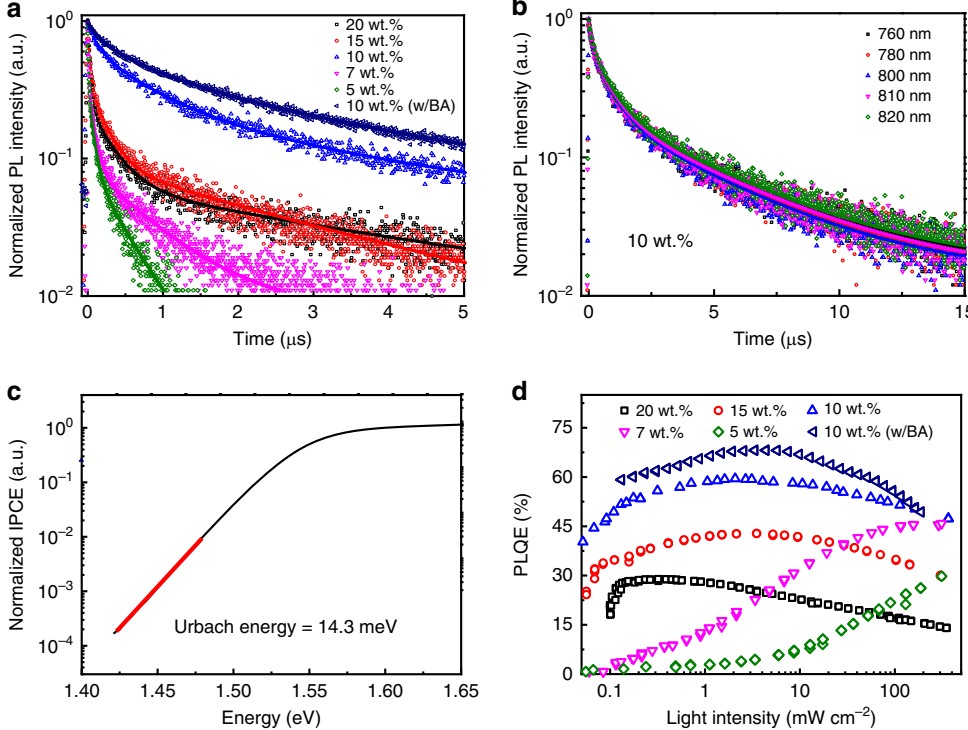

**Fig. 4** Optical properties of perovskite films. **a** Time-resolved PL for the FAPbI$_3$ films fabricated with different concentrations of precursor solutions under a fluence of 4 nJ cm$^{-2}$. **b** Time-resolved PL of 10 wt.% FAPbI$_3$ film at various emission wavelengths, which are almost identical. **c** Normalized IPCE at the absorption onset for a device fabricated with 10 wt.% FAPbI$_3$ film, measured by using FTPS. An $E_u$ of 14.3 meV can be calculated, as indicated by the red line. **d** Excitation-intensity-dependent PLQE of FAPbI$_3$ films fabricated from precursor solutions with different concentrations. The maximum PLQEs for 20 wt.%, 15 wt.%, 10 wt.%, 7 wt.%, 5 wt.% and BA treated 10 wt.% perovskite films are 29, 42, 60, 46, 30, and 68%, respectively

low defect levels associated with the large crystallites inside the grains.

## Discussion

We demonstrated that solution-processed PeLEDs can achieve good stability, and higher energy conversion efficiency than organic LEDs and quantum-dot LEDs. More importantly, the PeLEDs showed negligible EQE droop at high current density up to 300 mA cm$^{-2}$. This fact is very different with organic LEDs and quantum-dot LEDs, with which EQE droops at around 10 mA cm$^{-2}$ when exciton induced quenching effect occurs[16,17]. Notably, with optical structure to enhance the outcoupling efficiency, we can expect that the device efficiency can be further enhanced[8,13]. Our work suggests that the PeLED is an attractive technology to achieve low-cost, large-size, high-efficiency and high-brightness electricity-to-light conversion for lighting and display applications.

## Methods

**Synthesis and materials preparation**. FAPbI$_3$ precursor solutions with different concentrations were prepared by dissolving FAI and PbI$_2$ with the molar ratios of 2:1 in DMF and stirring at 60 °C for 8 h in a nitrogen-filled glovebox.

**Device fabrication**. The electron transport layer, ZnO-PEIE, hole transport layer, TFB and top electrodes, MoO$_x$/Au were prepared as previous report[6]. Here the perovskite films were prepared by spin-coating the precursor solution onto the PEIE treated ZnO films, followed by annealing on a hot plate at 100 °C. During the spin-coating of perovskite film, 100 μl of chlorobenzene was dropped onto the film after delay time of 5 s from the start of spinning[14]. For PeLEDs with BA treatment, 30 μl of BA dissolved in chlorobenzene (1 vol.%) was spin-coated onto the annealed perovskite film. Then the BA treated perovskite film was annealed at 100 °C for 5 min to remove chlorobenzene.

**Characterization**. All perovskite LED devices were characterized by combination of a fiber integration sphere (FOIS-1) couple with a QE-6500 spectrometer and a Keithley 2400 source meter[6]. The devices were swept from zero bias to forward bias at a rate of 0.05 V s$^{-1}$.

The simultaneous measurements of PL and EL were obtained by combination of lock-in amplifier (SR830), electric-meter (Keithley 2000, Keithley 2400), and photodector (Thorlabs PDA100A)[21]. The devices were encapsulated by glasses with ultraviolet-curable resin. During the measurement, the excitation light intensity (425 nm laser) was chopped (930 Hz) and kept extremely low, around 0.2 mW cm$^{-2}$. The EL and PL signal can be separated by using lock-in amplifier. The atmosphere temperature is 18 °C and the relative humidity is 30%.

The PL and EL mapping measurements of devices without encapsulation were recorded in air by a WITec alpha 300 R confocal Raman microscope. For PL mapping, the devices were excited by a 633 nm laser with an intensity of 1 μW. The PL intensity maps are obtained before and after the biasing of devices. For EL test, the devices worked at a constant current density of 100 mA cm$^{-2}$.

HRTEM samples were prepared by using duel beam focused-ion beam (FIB) equipment (FEI Quata 3D FEG). First, a 1 μm of Pt was deposited upon the device in 2 min. Second, the HRTEM samples were cut from the devices by FIB with the current of 3 nA. Third, the HRTEM samples were fixed to the TEM copper mesh by depositing Pt on the edge of the samples with the FIB current of 50 pA. At last, the samples on the copper mesh were thinned to around 200 nm. After the preparation of the samples, we immediately transferred them into the HRTEM chamber. The HRTEM images were collected on a Tecnai G2 F20 microscope operated at 200 keV.

UV-vis absorbance spectra were recorded on a UV-vis spectrophotometer with an integrating sphere (Lambda 950, PerkinElmer). The time resolved fluorescence spectra of the perovskite films were obtained by using an Edinburgh Instruments (FLS920) spectrometer. For the time-resolved PL measurements, the perovskite films were excited by a 638 nm pulsed diode laser. A technique was used to obtain PLQE of perovskite films by combination of CW laser, optical fiber, spectrometer and integrating sphere[34].

The surface morphology of perovskite films was obtained by using a JEOL JSM-7800F SEM.

FTPS was measured combining a Fourier-transform infrared spectroscopy (FTIR, Vertex 70, Bruker Optics) with a low-noise current amplifier (SR570, Stanford Research Systems). The photocurrent produced by the devices was amplified and fed back into the external detector port of the FTIR. $E_u$ was calculated by the IPCE data measured by FTPS, with a fitting error of ±0.1 meV.

## Data availability

The data that support the finding of this study are available from the corresponding author upon reasonable request.

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

## Acknowledgements

This work is financially supported by the Joint Research Program between China and European Union (2016YFE0112000), the Major Research Plan of the National Natural Science Foundation of China (91733302), the National Basic Research Program of China-Fundamental Studies of Perovskite Solar Cells (2015CB932200), the Natural Science Foundation of Jiangsu Province, China (BK20150043, BK20150064, BK20180085), the National Key R&D Program of China (2016YFB0401600, 2017YFB0404500, 2018YFB0406704), the National Natural Science Foundation of China (11474164, 61875084, 61634001, 51522209, 91433204), the National Science Fund for Distinguished Young Scholars (61725502), the Major Program of Natural Science Research of Jiangsu Higher Education Institutions of China (18KJA510002), the Synergetic Innovation Center for Organic Electronics and Information Displays, the Natural Science Foundation of Zhejiang Province, China (LY17A040008). We thank G. Zhu for the assistance of the HRTEM measurement, S. Wu and K. Du for the assistance of preparing the cross-sectional samples and Y. Liu, Q. Tao for the HRTEM analysis.

## Author contributions

J.W. had the idea for and designed the experiments. J.W. and W.H. supervised the work. Y.M. and Y.K. carried out the device fabrication and characterizations and HRTEM measurement. W.Z., M.X. L.Z., Y.T., W.X. and R.L. conducted the optical measurements. R.Y. and Y.W. participated in device fabrication and characterizations. Y.C. measured SEM. Y.S., J.L. and H.H. carried out TCSPC characterizations. Y.M. and F.G. performed the FTPS measurement, J.W. wrote the first draft of the paper. N.W., Y.J., and W.H. participated in data analysis and provided major revisions. All authors discussed the results and commented on the paper.

## Additional information

**Competing interests:** The authors declare no competing interests.

