## [Peer Review File · Nature Communications]

Editorial Note: This manuscript has been previously reviewed at another journal that is not operating a transparent peer review scheme. This document only contains reviewer comments and rebuttal letters for versions considered at Nature Communications .

Reviewers' comments:

Reviewer #1 (Remarks to the Author):

The authors have overall improved their manuscript since I last reviewed the paper. There is greater clarity and less unnecessary hype around the claims. I think the work is still important as the stability is quite strong, even with respect to recent reports of higher EQE devices.

Nevertheless, I still struggle to see the scientific advancement in its current version over just trying an additive and it working. This is because the authors don't yet have a satisfactory analysis of the action of the BA. In my mind it is firstly likely a moisture blocking layer, helping stability and perhaps ions moving between layers. However, it is also providing some form of passivating effect (increasing luminescence yields), which hasn't been properly probed by the authors.

The time-resolved PL they show does not include a reference sample without BA (provided it can be stabilised), just different concentrations of the precursor solution. This would be critical to make conclusions about the action of the BA.

Furthermore, the comment about recombination is not correct and is therefore misleading.

'The results show that the films with higher concentrations show more mono-charge carrier recombination under low carrier densities and the 10 wt.% film shows more bimolecular recombination which is the origin of radiative recombination (Supplementary Fig. 9).'

The 10% sample shows no change with fluence. A bimolecular regime would have a decreasing 'lifetime' with increasing fluence. Therefore this conclusion can not be drawn. The other samples show some weak increasing lifetime with fluence, likely related to some trap-filling. I'm not sure the authors can say much about it being monomolecular either. I think this is primarily because it is only measured over a very narrow fluence regime (corresponding to charge densities orders of magnitude lower than in an operating LED at peak EQE). A better analysis of the time-resolved kinetics including a reference and wider fluence range would improve the understanding of the enhancements.

Reviewer #2 (Remarks to the Author):

Authors present a high-efficiency pLED based on FAPbI₃. Using benzylamine treatment, they could obtain a T50 of 24 h, which is impressive. Compared to the previous version I have been reviewing, authors improved the paper to make it more suitable to be published in Nat. Comm. I added a few technical comments to the pdf, which authors might want to address. Regarding the style, it would be better to move some figures, which are discussed in detail in the main manuscript, from the SI to the main manuscript.

Reviewer #3 (Remarks to the Author):

The authors have satisfactorily addressed the reviewer concerns, and the paper is considerably improved. I feel that issues linger about film formation - both for the iodide and bromide FA perovskites, which are likely extrinsic and not intrinsic - but this need not hinder publication.

**Stable and bright formamidinium-based perovskite light-emitting diodes with**
**high energy conversion efficiency**

Yanfeng Miao¹, You Ke¹, Nana Wang¹, Wei Zou¹, Mengmeng Xu¹, Yu Cao¹, Yan
Sun¹, Rong Yang¹, Ying Wang¹, Renzhi Li¹, Jing Li², Haiping He², Yizheng Jin³,
Feng Gao⁴, Jianpu Wang^{1*} and Wei Huang^{1,5,6*}

¹*Key Laboratory of Flexible Electronics (KLOFE) & Institute of Advanced Materials*
*(IAM), Jiangsu National Synergetic Innovation Center for Advanced Materials*
*(SICAM), Nanjing Tech University (NanjingTech), 30 South Puzhu Road, Nanjing*
*211816, China.*

²*State Key Laboratory of Silicon Materials, School of Materials Science and*
*Engineering, Zhejiang University, Hangzhou 310027, China.*

³*Center for Chemistry of High-Performance and Novel Materials, State Key*
*Laboratory of Silicon Materials, and Department of Chemistry, Zhejiang University,*
*Hangzhou 310027, China.*

⁴*Biomolecular and Organic Electronics, IFM, Linköping University, Linköping 58183,*
*Sweden.*

⁵*Key Laboratory for Organic Electronics and Information Displays & Institute of*
*Advanced Materials (IAM), Jiangsu National Synergetic Innovation Center for*
*Advanced Materials (SICAM), Nanjing University of Posts & Telecommunications, 9*
*Wenyuan Road, Nanjing 210023, China.*

⁶*Shaanxi Institute of Flexible Electronics (SIFE), Northwestern Polytechnical*
*University (NPU), 127 West Youyi Road, Xi'an 710072, China.*

**Correspondence to: iamjpwang@njtech.edu.cn, iamwhuang@nwpu.edu.cn.*

Solution-processable organic lead halide perovskites show high
photoluminescence quantum efficiencies (PLQEs) and good charge transport,
making them attractive for low-cost light-emitting diodes (LEDs) with high energy
conversion efficiencies (ECEs). Despite recent advances in device efficiency, the
stability of perovskite LEDs (PeLEDs) is still a major obstacle. Here, we
demonstrate stable and bright PeLEDs with high ECEs by optimizing pure
$\text{NH}_2\text{CH}=\text{NH}_2\text{PbI}_3$ (FAPbI₃) films. Our PeLEDs show a peak ECE of up to 10.7%,
and a peak external quantum efficiency (EQE) of 14.2%, which is remarkably
improved compared to other FAPbI₃ PeLEDs without outcoupling enhancement.

[revised manuscript text omitted]

**Additional information**

Supplementary information is available in the online version of the paper. Reprints and
permissions information is available online at www.nature.com/reprints.

Correspondence and requests for materials should be addressed to J.W. and W.H.

**Competing financial interests**

The authors declare no competing financial interests.

**Figure 1. PeLED device structure and optoelectronic characteristics.** **a**, Device
 structure and cross-sectional HRTEM image (scale bar, 20 nm). **b**, Device EL spectra
 **upon various biases.** **c**, Current density and radiance versus driving voltage for the
 highest EQE device (10 wt.%). Radiance of $241 \text{ W sr}^{-1} \text{ m}^{-2}$ is obtained under 2.75 V. **d**,
 EQE and ECE versus current density for the highest EQE device (10 wt.%). A peak
 EQE of 14.2% is achieved at a current density of 188 mA cm^{-2} and a peak ECE of
 10.7% is obtained at a current density of 54 mA cm^{-2} . **e**, Histogram of peak EQEs
 measured from 82 devices, which shows an average peak EQE of 10.5% with a relative
 standard deviation of 16%.

**Figure 2. Optoelectronic characteristics of the aged devices. a,** EL and PL decay of
 the 3D PeLEDs without BA treatment at a constant current density of 100 mA cm⁻². **b,**
 EL and PL decay of the 3D PeLEDs with BA treatment at a constant current density of
 100 mA cm⁻². **c,** Normalized device EL, current density and EQE under cyclic voltages
 between -1 and 2.5 V after aging test. The EL/current density/EQE show recovery.

**Figure 3. SEM images of FAPbI₃ films fabricated with different concentrations of**
**precursor solutions (scale bar, 1 μm).**

**Figure 4. Optical properties of perovskite films.** **a**, Time-resolved PL for the FAPbI₃
 films fabricated with different concentrations of precursor solutions. **b**, Time-resolved
 PL of 10 wt.% FAPbI₃ film at various emission wavelengths, which are almost identical.
 **c**, Normalized IPCE at the absorption onset for a device fabricated with 10 wt.%
 FAPbI₃ film, measured by using FTPS. An E_u of 14.3 meV can be calculated, as
 indicated by the red line. **d**, Excitation-intensity-dependent PLQE of FAPbI₃ films
 fabricated from precursor solutions with different concentrations. The maximum
 PLQEs for 20 wt.%, 15 wt.%, 10 wt.%, 7 wt.% and 5 wt.% perovskite films are 29%,
 42%, 60%, 46% and 30% respectively.

Point-by-Point Response to Referees

Reviewer #1 (Remarks to the Author):

Comment #1: The authors have overall improved their manuscript since I last reviewed the paper. There is greater clarity and less unnecessary hype around the claims. I think the work is still important as the stability is quite strong, even with respect to recent reports of higher EQE devices. Nevertheless, I still struggle to see the scientific advancement in its current version over just trying an additive and it working. This is because the authors don't yet have a satisfactory analysis of the action of the BA. In my mind it is firstly likely a moisture blocking layer, helping stability and perhaps ions moving between layers. However, it is also providing some form of passivating effect (increasing luminescence yields), which hasn't been properly probed by the authors. The time-resolved PL they show does not include a reference sample without BA (provided it can be stabilised), just different concentrations of the precursor solution. This would be critical to make conclusions about the action of the BA.

Response: We thank the reviewer for recognizing the importance of our work and for the constructive comments. We have performed the excitation-intensity-dependent PLQE and PL lifetime measurements of the whole set of the perovskite films (**Figure 4**). Especially the PL lifetime was measured under a much wider fluence regime (**Supplementary Figure 9**). The results show that the BA treated 10 wt.% film has higher PLQE and longer lifetime than the untreated film at low excitations, which is consistent with the passivation effect of the BA treatment. We have added this in the revised manuscript (Line 197 to 200, Page 8, highlighted, Figure 4, Supplementary Figure 9).

Figure 4. Optical properties of perovskite films. **a**, Time-resolved PL for the FAPbI₃ films fabricated with different concentrations of precursor solutions under a fluence of 4 nJ cm^{-2} . **d**, Excitation-intensity-dependent PLQE of FAPbI₃ films fabricated from precursor solutions with different concentrations. The maximum PLQEs for 20 wt.%, 15 wt.%, 10 wt.%, 7 wt.%, 5 wt.% and BA treated 10 wt.% perovskite films are 29%, 42%, 60%, 46%, 30% and 68% respectively.

Supplementary Figure 9. Time-resolved PL of perovskite films under various excitation fluence. The films are fabricated with different concentrations of precursor solutions. **a**, 20 wt.%. **b**, 15 wt.%. **c**, 10 wt.%. **d**, 7 wt.% **e**, 5 wt.%. **f**, 10 wt.% with BA treatment.

Comment #2: Furthermore, the comment about recombination is not correct and is therefore misleading. The results show that the films with higher concentrations show

more mono-charge carrier recombination under low carrier densities and the 10 wt.% film shows more bimolecular recombination which is the origin of radiative recombination (Supplementary Fig. 9). The 10% sample shows no change with fluence. A bimolecular regime would have a decreasing 'lifetime' with increasing fluence. Therefore this conclusion can not be drawn. The other samples show some weak increasing lifetime with fluence, likely related to some trap-filling. I'm not sure the authors can say much about it being monomolecular either. I think this is primarily because it is only measured over a very narrow fluence regime (corresponding to charge densities orders of magnitude lower than in an operating LED at peak EQE). A better analysis of the time-resolved kinetics including a reference and wider fluence range would improve the understanding of the enhancements.

Response: We thank the reviewer for this useful comment. As shown in **Supplementary Figure 9**, all the samples with different concentrations have decreasing lifetimes with increasing fluence, and the 10 wt.% film has the longest lifetime under the low fluence. Together with the PLQE data, we can conclude that the 10 wt.% film has less trap-assisted non-radiative recombination (Line 195 to 197, Page 8, highlighted, Supplementary Figure 9).

Reviewer #2 (Remarks to the Author):

Comment #1: Authors present a high-efficiency PLED based on FAPbI₃. Using benzylamine treatment, they could obtain a T₅₀ of 24 h, which is impressive. Compared to the previous version I have been reviewing, authors improved the paper to make it more suitable to be published in Nat. Comm. I added a few technical comments to the pdf, which authors might want to address. Regarding the style, it would be better to move some figures, which are discussed in detail in the main manuscript, from the SI to the main manuscript.

Response: We thank the reviewer for well appreciating our work. We have moved Supplementary Figure 6d (Peak EQE distribution for devices with different concentrations) to the main manuscript (**Figure 3f**).

Comment #2: Line36: “at high excitations” you mean driving currents?

Response: The “high excitations” means high current density (≥ 300 mA cm⁻²). We have revised it as “high current densities” (Line 35, Page 2, highlighted).

Comment #3: Line 66: “multiplayer”

Response: We have corrected the word (Line 66, Page 3, highlighted).

Comment #4: Line 80: “unchanged upon various biases.” would be good to show normalized spectra maybe in the SI.

Response: We have added the normalized EL spectra in **Supplementary Figure 2a**.

Supplementary Figure 2. a, Normalized device EL spectra upon various biases.

Comment #5: Line 127: “the PL decline is slower than the EL decline” to me they are almost the same. I would be careful to draw big conclusions here.

Response: We think maybe the fluctuation of the PL data makes the PL change not so obvious. We have smoothed the PL data in **Figure 2b** and we believe now it is much more clearer than before.

Figure 2. Optoelectronic characteristics of the aged devices. b, EL and PL decay of the 3D PeLEDs with BA treatment at a constant current density of 100 mA cm^{-2} .

Comment #6: Line 181: “Eu measurement” Does the Eu really change for different concentrations? Usually, PLQE can change a lot due to non radiative recombination but Eu is not affected. The SI Fig. 8 does not look very reliable to me. A more sensitive measurement is required as e.g. the fit of the black curve is strongly affected by the few noisy points. Disregarding those point, Eu might be the same as for the 10%.

Response: We thank the reviewer for this useful comment. After removing those noisy points, the E_u values of higher concentration films are still larger than the 10 wt.% film, as shown in below, **Supplementary Figure 8**.

Supplementary Figure 8. Normalized IPCE at the absorption onset for devices fabricated with 20 wt.%, 15 wt.%, and 10 wt.% FAPbI₃ films, measured by using FTPS.

Comment #7: Line 182: “highly energetically ordered with a very low level of defect density.” this is not so convincing to me. There could be other reasons that there are some defects formed which are not related to energetical order.

Response: We agree that there are some defects regardless with energetic order. We now modify the sentence as: The transient PL, PLQE and E_u measurement results together suggest that the fabricated 10 wt.% FAPbI₃ perovskite thin film is highly energetically ordered with a very low level of defect density (Line 187-190, Page 8, highlighted).

Comment #8: Line 189: “The results show that” I cannot see this. In my eyes, there is hardly any difference in the fluence dependence in S9. Please explain more.

Response: We thank the reviewer for this comment which is similar to the comment of Reivewer #1. We have measured the time-resolved PL of perovskite films under a much wider fluence regime (**Supplementary Figure 9**). It shows that all the samples with different concentrations have decreasing lifetimes with increasing fluence, and the 10 wt.% film has the longest lifetime at low fluence. Together with the high PLQE of the 10 wt.% film (**Figure 4d**), those results suggest less trap assisted non-radiative recombination with the 10 wt.% film than other films. (Line 195 to 197, Page 8, highlighted, Supplementary Figure 9).

Comment #9: Line 190: “more mono-charge” what is this?

Response: We have modified the description of the fluence dependent PL lifetime (Line 195 to 197, Page 8, highlighted).

Reviewer #3 (Remarks to the Author):

Comment #1: The authors have satisfactorily addressed the reviewer concerns, and the paper is considerably improved. I feel that issues linger about film formation - both for the iodide and bromide FA perovskites, which are likely extrinsic and not intrinsic - but this need not hinder publication.

Response: We thank the reviewer for the positive comment.

Reviewers' comments:

Reviewer #1 (Remarks to the Author):

I am satisfied that the suggestions have been satisfactorily addressed. I think the manuscript now has a sound scientific/photophysical platform to understand the impressive device performance and stability. So I can recommend the manuscript for publication as a strong addition to the PeLED community.

Reviewer #2 (Remarks to the Author):

First I got confused that I should review this paper because I thought it had been published in Nat. Photonics in March: "Rational molecular passivation for high-performance perovskite light-emitting diodes". Now I see that these are two different studies by almost the same authors. There are differences in the passivation molecule used etc., however, the system FAPbI₃ is the same and the performance reported in the Nat. Photonics paper is higher. Therefore, I have high doubt recommending publication in Nature Comm. For the moment I recommend to reject the paper straight away because it does not even refer to the Nature Photonics. Prior to a potential detailed re-review, authors need to make very clear how this submission refers to their other paper. E.g. statements regarding the highest ECEs are already falsified by the authors themselves.

Point-by-Point Response to Referees

Reviewer #1 (Remarks to the Author):

Comment #1: I am satisfied that the suggestions have been satisfactorily addressed. I think the manuscript now has a sound scientific/photophysical platform to understand the impressive device performance and stability. So I can recommend the manuscript for publication as a strong addition to the PeLED community.

Response: We thank the reviewer for the positive comment.

Reviewer #2 (Remarks to the Author):

Comment #1: First I got confused that I should review this paper because I thought it had been published in Nat. Photonics in March: "Rational molecular passivation for high-performance perovskite light-emitting diodes". Now I see that these are two different studies by almost the same authors. There are differences in the passivation molecule used etc., however, the system FAPbI₃ is the same and the performance reported in the Nat. Photonics paper is higher. Therefore, I have high doubt recommending publication in Nature Comm. For the moment I recommend to reject the paper straight away because it does not even refer to the Nature Photonics. Prior to a potential detailed re-review, authors need to make very clear how this submission refers to their other paper. E.g. statements regarding the highest ECEs are already falsified by the authors themselves.

Response: We thank the reviewer for this comment.

Firstly, these two papers report different strategies to improve the performance of perovskite LEDs. In this manuscript, we mainly discuss the influence of precursor solution concentration on the morphology and trap density of pure 3D FAPbI₃ perovskite. Moreover, we introduce a benzylamine (BA) interface layer through a simple post-deposition treatment of perovskite films, which can passivate the interfacial defects and enhance the moisture resistance. However, the Nature Photonics paper focuses on the effect of additives on device performance, more specifically, how the hydrogen-bond of the additives affect the device performance of perovskite LEDs.

Secondly, the ECE in this work is still higher than that in other work as we have claimed in the manuscript, which is compared to that of device without outcoupling enhancement (Page 2, Line 32 and Page 4, Line 91). Actually the high efficiency of the Nature Photonics paper is partially due to the enhanced light outcoupling caused by nano-island feature, as shown in the TEM images. To refer to the Nature Photonics paper, we have added it in the reference (Page 3, Line 54-56, highlighted).

In addition, the device in this work exhibits outstanding lifetime, which is longer than

that in the Nature Photonics paper even under 4 times higher current densities. In order to further illustrate the mechanism of the stability enhancement, we have performed the PL and EL mapping of the devices by using a confocal fluorescence microscope. We find that the improved lifetime of the device is mainly due to the reduced interfacial defects and enhanced moisture resistance of the perovskite film after BA treatment (Page 6-7, Line 145-160, highlighted).

Reviewers' comments:

Reviewer #2 (Remarks to the Author):

For me the innovation of this work compared to the Nature Photonics remains rather unclear as "simply controlling the concentration...", as authors write in the introduction, is a standard approach and the optimum concentration should have been chosen for the Nat Photonics paper. The main message of this paper are the high values, see the abstract. Maybe the abstract should mention somehow on the innovation of this work and how these high values have been achieved.

Some measurements are not that convincing to me, e.g. the PL vs. EL. In Fig. 2. The PL is very noisy, in a, maybe explaining why it does not follow the EL.

"Eu of 14.9 meV for the 10 wt.% perovskite film and 14.3 meV" Is the accuracy of this measurement so high that 0.1 meV can be resolved?

The added mapping does not add much insights. Authors say that degradation mainly happens from the contacts and due to moisture. However, the EQE stability data is shown for encapsulated devices where moisture should not play a role. Therefore, the degradation in the two cases (inert and ambient atmosphere) should be different and the implications for application are enormous as the latter can be avoided by encapsulation whereas the former cannot.

These points should be clarified prior to publication

Point-by-Point Response to Referees

Reviewer #2 (Remarks to the Author):

Comment #1: For me the innovation of this work compared to the Nature Photonics remains rather unclear as “simply controlling the concentration...”, as authors write in the introduction, is a standard approach and the optimum concentration should have been chosen for the Nat Photonics paper. The main message of this paper are the high values, see the abstract. Maybe the abstract should mention somehow on the innovation of this work and how these high values have been achieved.

Response: We thank the reviewer for recognizing the importance of our work and for the constructive comments. We have added this in the revised manuscript (Line 33, Page 2, highlighted).

Comment #2: Some measurements are not that convincing to me, e.g. the PL vs. EL. In Fig. 2. The PL is very noisy, in a, maybe explaining why it does not follow the EL.

Response: In order to avoid the potential perturbation caused by excitation light, we measure the PL signal under a low excitation ($\sim 0.2 \text{ mW cm}^{-2}$) which makes the PL data noisier than the EL data. However, the below figure shows that the real PL data only has a small fluctuation ($< 4\%$) around the averaged PL curve. So we believe the comparison of the PL/EL is reliable.

Comment #3: "Eu of 14.9 meV for the 10 wt.% perovskite film and 14.3 meV" Is the accuracy of this measurement so high that 0.1 meV can be resolved?

Response: We thank the reviewer for this useful comment. By fitting the FTPS data, the E_u with an accuracy (standard error) of ± 0.1 meV can be obtained. We have added this in the manuscript (Line 300-301, Page 12 highlighted).

Comment #4: The added mapping does not add much insights. Authors say that degradation mainly happens from the contacts and due to moisture. However, the EQE stability data is shown for encapsulated devices where moisture should not play a role. Therefore, the degradation in the two cases (inert and ambient atmosphere) should be different and the implications for application are enormous as the latter can be avoided by encapsulation whereas the former cannot.

Response: We thank the reviewer for this comment. The comparison of devices with and without BA treatment shows that the BA treated device exhibits improved lifetime in air due to the reduced interfacial defects and enhanced moisture resistant. This will cause similar degradation process for encapsulated device due to the slow penetration of moisture. Especially, in our lab we only encapsulate the device with simple ultraviolet-curable resin.

REVIEWERS' COMMENTS:

Reviewer #2 (Remarks to the Author):

As already said in my previous report, the paper is ok for Nat Com. Also as detailed before, my opinion remains that some of the explanations are not supported by sufficient experimental evidence. E.g.:

Authors claim that they can deduce an Urbach energy with an error lower than 0.1 meV to be able to claim meaning to measured differences of 0.1 meV. I don't think this is possible because of

- measurement inaccuracy
- fitting error
- ambiguity on selecting the energy range to fit
- the strange signal below 1.42 V
- sample to sample variations

I think authors would need to analyze their data much more carefully including statistics, a proper analysis of their fit errors etc.

Point-by-Point Response to Referees

Reviewer #2 (Remarks to the Author):

Comment #1: As already said in my previous report, the paper is ok for Nat Com. Also as detailed before, my opinion remains that some of the explanations are not supported by sufficient experimental evidence. E.g.: Authors claim that they can deduce an Urbach energy with an error lower than 0.1 meV to be able to claim meaning to measured differences of 0.1 meV. I don't think this is possible because of

- measurement inaccuracy
- fitting error
- ambiguity on selecting the energy range to fit
- the strange signal below 1.42 V
- sample to sample variations

I think authors would need to analyze their data much more carefully including statistics, a proper analysis of their fit errors etc.

Response: We thank the reviewer for the positive comment. The signal below 1.42 eV is the noisy signal. We have removed the noisy signal and fitted the Urbach energy within the clear edge of the FTPS data. The error of ± 0.1 meV is the fitting error. We have added this in the revised manuscript (Line 300, Page 12).